# An Advanced Decision Tree-Based Deep Neural Network in Nonlinear Data Classification

Mohammad Arifuzzaman [1], Md. Rakibul Hasan [1], Tasnia Jahan Toma [2], Samia Binta Hassan [2] and Anup Kumar Paul [1,*]

[1] Department of Electronics and Communications Engineering, East West University, Dhaka 1212, Bangladesh
[2] Department of Computer Science and Engineering, East West University, Dhaka 1212, Bangladesh
* Correspondence: anuppaul@ewubd.edu

**Abstract:** Deep neural networks (DNNs), the integration of neural networks (NNs) and deep learning (DL), have proven highly efficient in executing numerous complex tasks, such as data and image classification. Because the multilayer in a nonlinearly separable data structure is not transparent, it is critical to develop a specific data classification model from a new and unexpected dataset. In this paper, we propose a novel approach using the concepts of DNN and decision tree (DT) for classifying nonlinear data. We first developed a decision tree-based neural network (DTBNN) model. Next, we extend our model to a decision tree-based deep neural network (DTBDNN), in which the multiple hidden layers in DNN are utilized. Using DNN, the DTBDNN model achieved higher accuracy compared to the related and relevant approaches. Our proposal achieves the optimal trainable weights and bias to build an efficient model for nonlinear data classification by combining the benefits of DT and NN. By conducting in-depth performance evaluations, we demonstrate the effectiveness and feasibility of the proposal by achieving good accuracy over different datasets.

**Keywords:** neural network; deep neural network; decision tree; nonlinear data classification; back propagation; gradient descent





## 1. Introduction

When sufficient training data and computing power are available, one of the consolidated findings of contemporary (very) deep-learning approaches [1–4] is that their joint and unified method of learning feature representations together with their classifiers significantly outperforms traditional feature descriptors and classifier pipelines.

Learning from large datasets is now a necessity in many sectors, including machine learning, pattern identification, medical diagnosis, speech recognition, localization, cybersecurity, and image processing, thanks to advancements in science and technology [5–11]. Decision tree learning benefits from easy implementation, few parameters, low calculation, and the ability to adapt to different huge data types. In decision trees, the scale of the tree somewhat reflects the degree of generalizability. The rules retrieved from the tree become more complex as the tree's scale increases. Overfitting issues will result from overly complex rules [12]. Making the optimum decision tree as compact as feasible is crucial without compromising classification accuracy. Neural networks have been demonstrated to be a successful learning technique for carrying out classification tasks, particularly when high-dimensional data are input and the relationship between the input and output is complex [13]. According to studies, the depth of neural network models improves the classification or prediction accuracy by exponentially increasing their ability to represent data. However, a lot of training time will be needed for this process.

Numerous ensemble learning techniques about neural networks and decision trees have been put out by academics in recent years. The author of [14] suggested using a neural network to preprocess each attribute's relationship with the target attribute and then create

a derivative relationship between each attribute and the classification outcome to create a tree. However, the algorithm's time complexity is significant. The author of [15] proposed a hybrid learning model of the BP algorithm based on the C4.5 algorithm and optimization to address the issue of difficult input parameter selection for the BP neural network and hidden layer nodes. However, because the model is a binary tree, it is unable to address the multi-classification issue. An extreme learning machine tree (ELM-Tree) model was proposed in [16], although the technique leverages information gain in node splitting, which has a tendency to be biased towards the attributes of picking more branches and results in overfitting.

In recent years, deep learning has become one of the breakthroughs in the field of machine learning [17]. In deep learning, deep neural network (DNN), developed from the neural network (NN), is a machine learning technique imitating the human nervous system and the brain's structure [18–21]. In general, NN consists of the input layer, hidden layer, and output layer [22,23], where each node or unit is interconnected to its peer entities in the adjacent layer, and the corresponding weight values are introduced in every connection [5]. DNNs are widely used to solve various problems, including automated image classification, data classification, data clustering, data approximation, data optimization, computer vision application, natural language processing, and predictive analysis [7,21,24–30]. DNN is also proven to be a cogent method for solving large-scale real-world problems [31].

Moreover, decision tree (DT) models are widely used for classification, where they perform a recursive partition for the input data and assign a weight to the final node. One of the critical advantages of DT models is that they are simple to decipher. Further, DT-based models are comparatively similar and, in some cases, better than NNs at predicting or classifying when using tabular data [32].

Nonlinear data classification, namely planar data classification, which involves multiple classes in the real-world [33–35], is a crucial research theme in the data classification field. In this context, classification is one of the most important aspects in a variety of practice scenarios where it plays an important role, such as environmental monitoring, multi-colored classification of space data, including stars, mars, the moon, or any complex data pattern, urbanization, disaster-affected areas, and traffic supervision [36]. Different neural network models have been proposed to segment or cluster a dataset [37]. In general, logistic regression is mostly used for linearly separable data since it gives a lower classification error [38]. In this paper, we use nonlinear separable complex data to address various practical scenarios where a single decision tree or logistic regression demonstrates a relatively high classification error rate. The NN model can automatically learn from complex data, which may contain millions of data points or thousands of parameters in a dataset [22].

To enable a considerable performance enhancement in nonlinear data classification, we propose the integrated models of DT and DNN for nonlinear data classification; namely decision tree-based deep neural network (DTBDNN). The proposal then realizes a better solution to the problem of nonlinear data with complex and low-contrast objects. While it would be quite difficult for the traditional algorithm to classify nonlinearly separable data [38], our proposal can effectively resolve the speculation and decipher capacity. Better still, the proposed DTBDNN model is developed using DT, in which we used a back propagation algorithm along with a gradient descent optimizer to optimize the trainable parameters. Second, we do not restrict the decision tree split to being binary; rather, we used a differentiable soft-binning [39] function to split nodes into multiple (>2) leaves that further improve the performance of the DTBDNN model.

The rest of the paper is organized as follows: We start with related work in Section 2. Section 3 describes the materials and methods of our proposed model. Section 4 depicts the results of our proposed model, where we made an analysis of the results, and finally, we conclude in Section 5.

## 2. Related Work

### 2.1. Background on Decision Tree

J. Ross Quinlan, a machine learning researcher, created the ID3 (Iterative Dichotomiser) decision tree method in the late 1970s and early 1980s. E. B. Hunt, J. Marin, and P. T. Stone's earlier study on concept learning systems were expanded upon in this paper. Later, Quinlan presented C4.5 (a replacement for ID3), which went on to serve as a standard by which newer supervised learning algorithms are frequently measured. The creation of binary decision trees was covered in the 1984 book Classification and Regression Trees (CART), written by a team of statisticians that included L. Breiman, J. Friedman, R. Olshen, and C. Stone. Though they were developed independently at about the same time, ID3 and CART use a similar method to learn decision trees from training tuples. An explosion of research on decision tree induction was spurred by these two cornerstone techniques [40].

The way the attributes are chosen when building the tree is one of the differences between decision tree algorithms. A heuristic for choosing the splitting criterion that "best" divides a given data partition, D, of class-labeled training tuples into distinct classes is known as an attribute selection measure. The ideal partition would be pure (i.e., all the tuples that fall into a given partition would belong to the same class) if we were to divide D into smaller partitions based on the results of the splitting criterion. The splitting criterion that yields the closest results in such a case is conceptually the "best" splitting criterion. Because they specify how the tuples at a specific node are to be split, attribute selection measures are also known as splitting rules.

Each attribute describing the given training tuples is ranked using the attribute selection measure (The best result is determined by the measure's highest or lowest score (i.e., some measures strive to maximize while others strive to minimize)). For the provided tuples, the attribute with the highest score for the measure is selected as the dividing attribute. A split point or a splitting subset must also be defined as part of the splitting criterion if the splitting attribute has continuous values or if binary trees are our only option. The splitting criterion is marked on the tree node made for partition D, branches are developed for each result of the criterion, and the tuples are partitioned as necessary. Three widely used attribute selection metrics are information gain, gain ratio, and Gini index.

**Information gain**: Information gain is the criterion used by ID3 to choose attributes. This measurement is based on Claude Shannon's ground-breaking information theory research, which examined the "information content" of signals. Let node N stand in for or contain the partition D tuples. The splitting attribute for node N is determined to be the one with the greatest information gain. This feature represents the least randomness or "impurity" in the generated partitions and reduces the amount of information required to categorize the tuples in those partitions. Such a method reduces the anticipated number of tests required to categorize a given tuple and ensures the discovery of a simple (but not necessarily the simplest) tree.

The expected information required to categorize a tuple in D is provided by

$$Info(D) = -\sum_{i=1}^{m} P_i log_2 P_i \qquad (1)$$

where $P_i$ is calculated as $|C_{i,D}|/|D|$ and represents the non-zero likelihood that each given tuple in D belongs to class $C_i$. The average amount of information required to determine a tuple's class label in D is called Info(D) or entropy of D.

Now, to categorize the tuples in D based on an attribute A that had v different values, such as $a_1, a_2, \cdots, a_v$, as seen in the training data. These values precisely equate to the v results of a test on A if A has discrete values. D can be divided into v divisions or subsets, $D_1, D_2, \cdots, D_v$, depending on the value of attribute A, where $D_j$ includes the tuples in

D that match the outcome $a_j$ of A. These divisions would line up with the branches that emerged from node N. This amount is measured by

$$Info_A(D) = \sum_{j=1}^{v} \frac{|D_j|}{|D|} \times Info(D_j) \tag{2}$$

The difference between the initial information requirement (i.e., based solely on the proportion of classes) and the new requirement (i.e., as determined after partitioning on A) is known as the information gain. That is,

$$Gain(A) = Info(D) - Info_A(D) \tag{3}$$

Gain(A) thus informs us of the gain that would result from branching on A. It is the anticipated decrease in the information needed to be brought on by understanding the value of A. The splitting attribute at node N is determined to be attribute A with the biggest information gain, Gain(A).

**Gain ratio**: The information gain metric favors tests with a wide range of results. In other words, it favors choosing qualities with a lot of possible values. Consider a property that serves as a distinctive identifier, such as a product ID. With a split based on product ID, there would be as many partitions as there are values, each carrying a single tuple. Each partition is pure; hence, the only data needed to categorize data set D using this partitioning would be $Info_{product_{ID}}(D) = 0$. As a result, partitioning on this attribute yields the most information. It is obvious that such a split is not useful for categorization.

In order to combat this prejudice, C4.5, the successor to ID3, introduces an addition to information gain known as a gain ratio. It uses a "split information" value defined analogously to Info(D) as a type of normalization to apply to information gain and is defined as

$$SplitInfo_A(D) = -\sum_{j=1}^{v} \frac{|D_j|}{|D|} \times log_2\left(\frac{|D_j|}{|D|}\right) \tag{4}$$

This value shows the potential information that might be produced by partitioning the training data set, D, into v groups, each grouping the results of a test on attribute A. Notably, it takes into account the proportion of tuples that have each outcome relative to the total number of tuples in D for each outcome. It is distinct from information gain, which evaluates the classification of newly acquired information based on the same partitioning. A definition of the gain ratio is

$$GainRatio(A) = \frac{Gain(A)}{SplitInfo_A(D)} \tag{5}$$

The attribute chosen as the splitting attribute is the one with the highest gain ratio.

**Gini Index**: In CART, the Gini index is employed. The Gini index calculates the impurity of D, a data partition or collection of training tuples as

$$Gini(D) = 1 - \sum_{i=1}^{m} P_i^2 \tag{6}$$

where $P_i$ is the probability that a tuple in D belongs to class $C_i$ and is estimated by $|C_{i,D}|/|D|$. The sum is computed over m classes.

For each attribute, the Gini index takes a binary split into account. We calculate a weighted total of the impurity of each resulting partition while considering a binary split. As an illustration, if D is partitioned into D1 and D2 by a binary split on A, D's Gini index after that partitioning is

$$Gini_A(D) = \frac{|D_1|}{|D|} Gini(D_1) + \frac{|D_2|}{|D|} Gini(D_2) \tag{7}$$

Each of the potential binary splits is taken into consideration for each attribute. The subset that has the lowest Gini index for a discrete-valued property is chosen as the subset's splitting subset.

Each potential split-point must be taken into account for continuous-valued attributes. Similar to the information gain approach previously discussed, the midpoint between each pair of (sorted) neighboring values is taken into consideration as a potential split-point. The split-point for a particular (continuous-valued) attribute is taken to be the point producing the smallest Gini index for that attribute. Remember that $D_1$ is the set of tuples in D satisfying the $A \leqslant SplitPoint$, and $D_2$ is the set of tuples in D satisfying the $A > splitPoint$, given a potential split-point of A.

The reduction in impurity that would result from a binary split on an attribute A with discrete or continuous values is

$$\Delta Gini(A) = Gini(D) - Gini_A(D) \tag{8}$$

The splitting attribute is chosen to optimize impurity reduction (or, equivalently, to have the lowest Gini index). The splitting criterion is the combination of this characteristic plus either its splitting subset (for a discrete-valued splitting attribute) or split-point (for a continuous-valued splitting attribute).

### 2.2. Neural Networks and Hybrid Models

Deep learning has surpassed human-level performance and capability in many areas, such as data classification, prediction and forecasting, the decision to approve loan applications, the time taken to deliver any object, etc. [41,42]. A decision tree creates a model that predicts the value of the targeted data or variable through the learning of simple decision rules from the data features. The DT algorithm is an easy one, as it is understandable and interpretable. DT works better for both categorical and numerical data and is able to handle multi-output data. In [43], the authors review several optimization methods with deep learning design, such as deep convolutional neural networks, recurrent neural networks, reinforcement learning, and autoencoders, to improve the accuracy of the training and show how we could reduce the training time with iterations.

Despite the enormous success of neural networks over the past decade, several industries, including health and security, have not adopted them widely or in a way that makes them more dependable. Researchers started looking into approaches to explain neural network decisions as a result of this fact. Saliency maps, approximation via interpretable methods, and joint models are some of the methodologies used to explain neural network judgments [44].

Saliency maps are a means to draw attention to the parts of the input that a neural network uses most frequently while making predictions. To show an input-specific linearization of the entire network, an earlier study [45] uses the gradient of the neural network output with respect to the input. Another piece of work [46] uses a deconvnet to return to decisions' features. These methods frequently produce noisy saliency maps that make it difficult to understand the choices that were made. The derivative of a neural network's output with respect to an activation, often the one just before completely connected layers, is used in another track of approaches [47–50]. These approaches lack a thorough logical justification for the decision while being beneficial for tasks such as determining whether the decision's backing is solid.

The conversion of interpretable by-design models, such as decision trees, to neural networks, has attracted attention. A technique for initializing neural networks with decision trees was developed in [51]. Decision tree equivalents for neural networks are also provided by [32,52,53]. These works' neural networks have particular topologies; hence, there is no generalization to any model. In [54], neural networks were trained so that trees could reasonably approximate their decision limits. Decision trees are only used as a regularization in this work; they are not provided as a correlation between neural networks and decision trees. In [55], a decision tree was trained using a neural network. This tree

distillation performs badly on the tasks that the neural network was trained on since it approximates a neural network rather than performing a direct conversion.

Joint neural network and decision tree models [56–62] typically employ deep learning to support some trees or to create a neural network structure that mimics a tree. In a recent paper [63], a decision tree is used in place of a neural network's final fully connected layer. Since neural networks' fundamental characteristics remain the same, an explanation is sought through the provision of a method for people to judge if a decision is good or bad rather than through a thorough, logical analysis of it.

In [64], the authors discuss the characteristics of DNN for image processing, and they provide two typical algorithms for saliency detection by using DNN. They then analyze three future robust developments of deep learning. The authors in [65] present a deep learning method for the machine identification of traffic signals. First, various stochastic gradient optimization algorithms, such as SGD (Stochastic Gradient Descent), nesterov accelerated gradient (NAG), RMSprop, and Adam, are tested. Subsequently, several configurations of spatial transformer networks are studied. A model with feature extraction and a learning algorithm of DNN is proposed in [66] to classify and recognize the patterns in Antarctica with hydrological features and is compared with some existing classification methods. The study in [67] proposed a two-stage deep feature fusion for scene classification, where the authors showed the advantage of using lower-layer features compared to exploiting fully connected layers.

Some commonly used deep learning architectures and their practical implementations are addressed in [41]. The authors surveyed four deep learning architectures, namely autoencoder, convolutional neural network, deep belief network, and restricted Boltzmann system, to provide an up-to-date overview. In this context, the authors in [32] proposed a deep neural decision tree (DNDT) by using the NN toolkit, and they evaluated the model's performance on various tabular datasets. In many datasets, they have proved that a decision tree-based neural network can achieve better accuracy compared to only a decision tree-based approach or only an NN-based approach. Another notable recent approach to constructing a deep forward neural network using a decision tree is introduced in [51], where the authors used their model to classify iris, digits, wine, and breast cancer data. However, their proposed model does not work for classifying non-planar data.

An extreme learning machine tree (ELM-Tree) model was proposed in [16]. The model tree provided in [68] and the ELM-Tree is comparable. A model tree and an ELM-Tree differ in that an ELM-Tree has each leaf node be an ELM, whereas a model tree is a decision tree with linear regression functions as leaf nodes. Single-hidden layer feedforward neural networks can be trained using ELMs or emergent learning methods. In an ELM, the output weights are calculated analytically using the pseudoinverse of the hidden layer output matrix, whereas the input weights are allocated at random [69–71]. In the ELM-Tree approach, a threshold is provided to decide whether or not to divide a node further. According to the class of impurity, if the learner chooses to stop splitting a node, it will either turn into a conventional leaf node or an ELM leaf node. Then, a parallel ELM-Tree model for big-data classification is created by parallelizing computation across five ELM-Tree components. Although the technique leverages information gain in node splitting, which has a tendency to be biased towards the attributes of picking more branches and results in overfitting.

## 3. Materials and Methods

The research methods and proposed models, along with their algorithms, have been discussed in this section as follows:

### 3.1. The Proposed DTBDNN Model

The relation between input and output data gets more complicated in the case of high-dimensional input data and a large number of training samples [36]. For a particular data classification test case, it is difficult to find how a single neural network predicts

a particular classification decision due to their dependency on distributed hierarchical representations [72]. Hence, in this research, we aim to build an efficient solution that acquires knowledge using a DNN model. This acquired knowledge is then expressed in another model that exploits the hierarchical decision tree structure to predict a particular classification decision efficiently and with good accuracy.

This section represents the proposed model, namely the decision tree-based deep neural network (DTBDNN) for nonlinear data classification that considers DNN with multiple hidden layers. In our model, we can route the input examples to leaf nodes of the neural network (the output layer) and classify them. Thus, training the neural network becomes training the soft bin cut points, and finally, leaf node classifiers perform the final computation for the classification decision. Then, we demonstrate a considerable difference between our proposed model and other state-of-the-art models when classifying nonlinear data. Further, the performance evaluations of the proposed DTBDNN in terms of accuracy and loss error are presented. The architectural overview of the methodology for building the proposed DTBDNN models is shown in Figure 1.

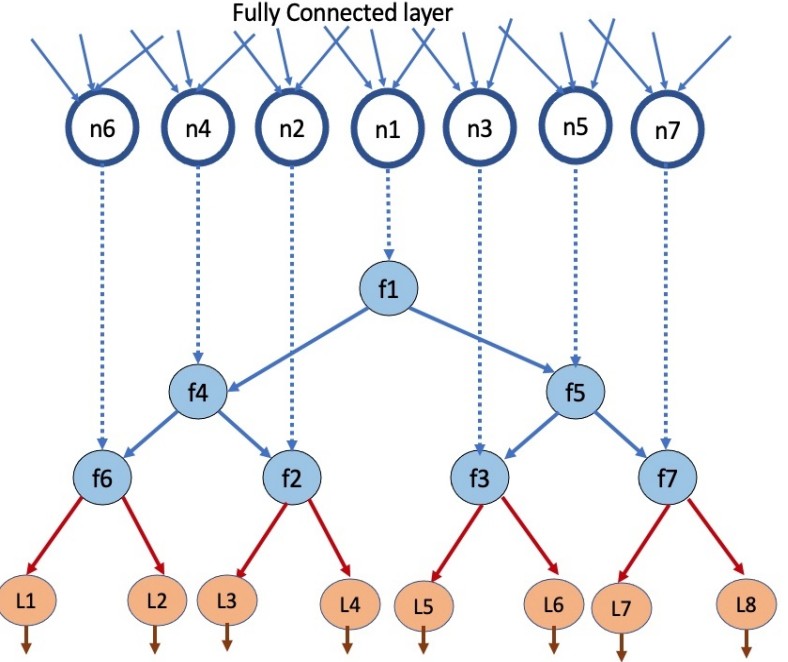

**Figure 1.** Architectural overview of the proposed model. Illustration of how to implement a decision tree-based neural network. The routing (split) decisions are created when each output of $f_n$ is brought into correspondence with a split node in a tree. The assignment of output units to decision nodes can be performed in any order (the one we show allows a simple visualization). As a result of resolving the convex optimization issue, the circles at the bottom are leaf nodes containing probability distributions over the multiclass classification problem.

We defined decision functions $f_n$ in terms of real-valued functions $f_n = tanh(W^T X + b)$, which are related but not necessarily independent due to the shared parametrization. By embedding functions $f_n$ within a deep neural network with parameter W, we hope to give the trees the ability to learn new features. In particular, each function $f_n$ can be viewed as a linear output unit of a deep network that will be converted into a routing choice by the action of $f_n$, which uses hyperbolic tangent activation to provide a response in the [−1, 1] range. A schematic illustration of this concept is shown in Figure 1, which demonstrates how decision nodes can be built using commonly available fully connected (or inner-product) and tanh layers in DNN frameworks. It is clear that the number of

output nodes in the fully connected layer above determines the number of split nodes. Because of this, the output units of the deep network under the proposed structure do not directly offer the final predictions, such as through a Softmax layer, but rather, each unit is in charge of influencing a node's decision inside the tree. In fact, a data sample x causes soft activations of the tree's routing decisions during the forward run through the deep network, which causes the routing function to generate a variety of leaf predictions that make up the final output.

The approach for determining if the datasets are linearly separable is depicted in the next section by using the data visualization technique. A dataset would be linearly separated if a linear function could separate the features of the dataset completely. In contrast, a nonlinear dataset is defined if no hyperplane lies on the pre-assigned side of the plane. For the DTBDNN models, we implemented a two-class NN classification with one hidden layer and a two-class DNN classification with multiple hidden layers, respectively. In the hidden layer(s) of NN and DNN, we use a nonlinear activation function unit, which is the tanh function for the forward propagation, whereas another activation function unit is used in a single node output layer, which is the sigmoid function. The reason for this is that deep neural networks' excellent speculation capacity is based on their use of conveyed representations in their hidden layers [73]. After tuning the performance of the NN and DNN models, we also found out and verified that the tanh activation function unit for every hidden layer would be best when we used the sigmoid activation function in the output layer. Further, we demonstrate that these nonlinear activation function combinations would be better than any other activation function combinations for any type of planar data classification.

### 3.1.1. System Model Overview

We design our decision tree-based deep neural network model by initially identifying the number of input, hidden, and output layers in the defined network structure. The main function that we used to make split decisions in our model is the soft-binning function. Typically, a soft-binning function takes the input features and produces an index of the bin to which the input features belong. Instead of using a hard-binning function, we have used a soft-binning function so that it can be differentiable during the back propagation phase of the neural network training. Then we construct multiple hidden layers. After that, we update the weights of the parameters and bias of the structure, where inputs are multiplied with the respective weights, adding a bias at each hidden node or unit, as shown in Figure 2. Typically, in each hidden unit, we have applied a nonlinear activation function "tanh", while in the output layer, the activation function undergoes a transformation based on another activation function, which is a sigmoid function. The input is squashed into a narrow output range from 0 to 1 and from −1 to 1 for the "sigmoid" and "tanh" functions, respectively. The acquired knowledge from the DNN model is expressed in another model that relies on the hierarchical decision tree algorithm to predict planar data classification with high accuracy. Then the prediction result of the final output layer is used as the solution to the problem of nonlinear data classification. For better presentation and exploration, we selected a nonlinear multi-colored flower dataset [74].

Figure 2 depicts the output of a given node as $H_i^{[L]}$, where "L" denotes the hidden layer numbers and "i" represents the specific units in that hidden layer. The output is calculated as the dot products of the input vector with the initialized pseudo-random weight (W) and adding the results with the bias (b). This intermediate result is then passed on to the nonlinear activation function $g$, which could be tanh, sigmoid, RELU, or Leaky RELU. We chose "tanh" as the activation function in a unit of the hidden layer because, by tuning the parameters, we found that it would gain better performance for the nonlinear data classification than using any other activation functions.

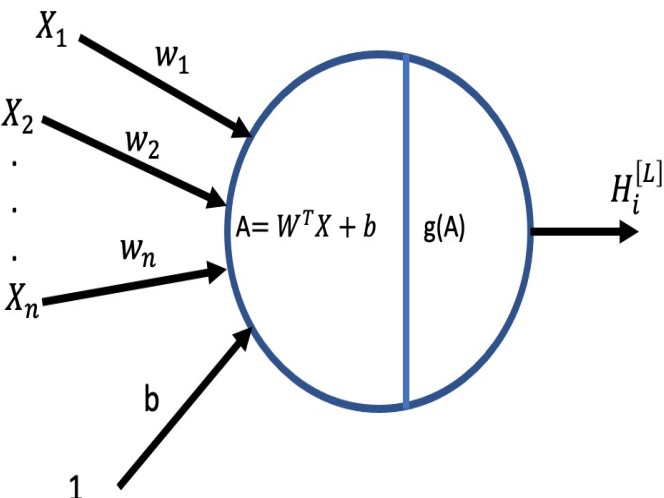

**Figure 2.** Hidden node in a hidden layer.

Figure 3 shows the general architecture of a deep neural network, where each node's functionality is depicted in Figure 2. The input to the network is an n-dimensional vector. The network contains L-1 hidden layers (two in this case) having n neurons each. Finally, there is one output layer containing k neurons (say, corresponding to k classes). Each neuron in the hidden layer and output layer can be split into two parts: preactivation and activation ($a_i$ and $h_i$ are vectors). The input layer can be called the 0-th layer, and the output layer can be called the *L*-th layer. $W_i \in \mathbb{R}^{n \times n}$ and $b_i \in \mathbb{R}^n$ are the weight matrix and bias vectors between layers $i-1$ and $i$ ($0 < i < L$). $W_L \in \mathbb{R}^{n \times k}$ and $b_L \in \mathbb{R}^k$ are the weight matrix and bias vectors between the last hidden layer and the output layer (L = 3 in this case). The preactivation at layer *i* is given by

$$a_i(x) = W_i h_{i-1}(x) + b_i \tag{9}$$

The activation at layer *i* is given by

$$h_i(x) = g(a_i(x)) \tag{10}$$

where *g* is called the activation function. The activation at the output layer is given by

$$h_L(x) = O(a_L(x)) \tag{11}$$

where *O* is the output activation function (softmax, linear, etc.). Therefore, for this three-layer network, as shown in Figure 3, the predicted output $\hat{y}$ is a linear combination of weights, inputs, and biases:

$$\hat{y}_i = O(W_3 g(W_2 g(W_1 x + b_1) + b_2) + b_3) \tag{12}$$

If the actual output is *y*, then we can calculate the loss/cost function depending on whether we want to solve the regression problem or a classification problem. For regression types of problems, the cost function is a mean square error and is defined as

$$J(\Theta) = min \frac{1}{N} \sum_{i=1}^{N} \sum_{j=1}^{k} (\hat{y}_{ij} - y_{ij})^2 \tag{13}$$

For classification types of problems, the cost function is a cross-entropy function and is defined as

$$J(\Theta) = -\frac{1}{N} (y_i log(\hat{y}_i) + (1 - y_i) log(1 - \hat{y}_i)) \tag{14}$$

where $\Theta = W_1, W_2, \cdots W_L, b_1, b_2, \cdots b_L$. In order to train the neural network, we have to minimize the cost function with respect to the parameters $\theta$ as follows

$$\Theta_{t+1} \leftarrow \Theta_t - \eta \nabla \Theta_t \tag{15}$$

where $\nabla \Theta_t = \left[ \frac{\delta J(\Theta)}{\delta W_t}, \frac{\delta J(\Theta)}{\delta b_t} \right]^T$, $t$ is the iteration index, and $\eta$ is the learning rate. The complete algorithm for training a deep neural network is given in Algorithm 1.

---

**Algorithm 1** Deep Learning Algorithm Forward Propagation Along With Gradient Descent.

---

**Require:** Network depth, L
**Require:** $W_i, i \in 1 \cdots L$, the weight matrices of the model
**Require:** $b_i, i \in 1 \cdots L$, the bias parameters of the model
**Require:** $X$, the input to process
**Require:** $y$, the target output
    $h_0 \leftarrow x$
    $t \leftarrow 0$
    $maxIterations \leftarrow 1000$
    $\Theta_0 = [w_0, b_0]$
    **while** $t + + < maxIterations$ **do**
        $k = 1$
        **while** $k \leq L$ **do**
            $a_k = W_k h_{k-1} + b_k$
            $h_k = g(a_k)$
            $k = k + 1$
        **end while**
        $\hat{y} = h_L$
        $J(\Theta) = \mathcal{L}(y, \hat{y})$          $\triangleright \mathcal{L}$ is the loss function
        $\Theta_{t+1} \leftarrow \Theta_t - \eta \nabla \Theta_t$          $\triangleright \nabla \Theta_t = \left[ \frac{\delta J(\Theta)}{\delta W_t}, \frac{\delta J(\Theta)}{\delta b_t} \right]^T$
    **end while**

---

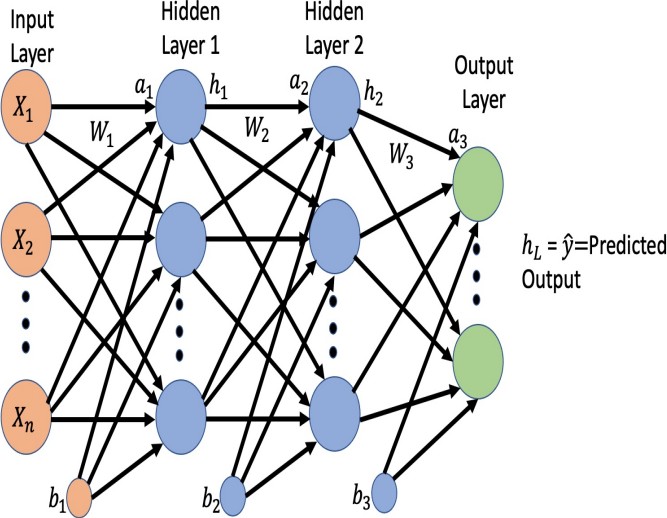

**Figure 3.** A Multilayer Deep Neural Network Architecture.

### 3.1.2. Decision Tree-Based Deep Neural Network (DTBDNN) Algorithm

In Algorithm 2, we introduce the algorithms for the DTBDNN model (number of hidden layers > 1). The main goal of this algorithm is to build a decision tree-based neural network framework in which the weight and bias are initialized and fed into the proposed DT with the optimized values from the DNN models. We set the parameter $n_x$ as the input

layer size, $n_h$ as the hidden layer size, and $n_p$ as the output layer size. Here, the parameters, the weight matrix of the hidden layer ($W_1$), and the weight matrix of the output layer ($W_2$) are initialized randomly to ensure that the initial weight cannot be large. Then, we initialize the bias vectors $b_1$ and $b_2$.

---

**Algorithm 2** Decision Tree-Based Deep Neural Network (DTBDNN) Algorithm.

---

**Require:** $X,W,b$, Input, Weight, bias of the model
**Require:** $y$, the target output (Binary or Multiclass)
**Require:** Initialization: $h_0 \leftarrow X$; $g_1 \leftarrow tanh$; $g_2 \leftarrow sigmoid$; $Cutpoints = [c_1, c_2, \cdots, c_n] \leftarrow$ SoftBinning(X,n); bias vector $b = [0, -c_1, -c_1 - c_2, \cdots, -c_1 - c_2 - \cdots - c_n]$

  **function ForwardPropagation**
  $i \leftarrow 1$
  **while** $i \leq L$ **do**                           $\triangleright$ $L$ is the total number of layers, here L = 2
      $a_i = W_i h_{i-1} + b_i$
      $h_i = g_i(a_i)$
      $i = i + 1$
      Cache $\leftarrow a_i, h_i$
  **end while**
  $\hat{y} \leftarrow h_L$
  **Return** $\hat{y}$, Cache
  **End ForwardPropagation**
  **function ComputeLoss**
  $J(W,b) = -\frac{1}{N}(y_i log(\hat{y}_i) + (1 - y_i)log(1 - \hat{y}_i))$
  **Return** $J(W,b)$
  **End ComputeLoss**
  **function BackPropagation**
  Import $a_1, h_1, a_2, h_2$ from Cache
  $g_1 \leftarrow 1 - g_1^2$
  $\delta a_2 \leftarrow h_2 - y$
  $\delta W_2 \leftarrow \frac{1}{m} \times (\delta W_2 \times h_1^T)$
  $\delta b_2 \leftarrow \frac{1}{m} \times \sum \delta a_2$
  $\delta a_1 \leftarrow W_2^T \times \delta a_2 \times g_1 \times a_1$
  $\delta W_1 \leftarrow \frac{1}{m} \times \delta a_1 \times X^T$
  $\delta b_1 \leftarrow \frac{1}{m} \times \sum \delta a_1$
  grads $\leftarrow [\delta W_1, \delta W_2, \delta b_1, \delta b_2]$
  **Return** grads
  **End BackPropagation**
  **function UpdateParameters**
  $i \leftarrow 1$
  **while** $i \leq L$ **do**
      $W_i \leftarrow W_i - \eta \times \delta W_i$
      $b_i \leftarrow b_i - \eta \times \delta b_i$
  **end while**
  Parameters$\leftarrow W_i, b_i$
  **Return** Parameters
  **End UpdateParameter**

---

Then we apply the "soft-binning" function [39] on the input x to split nodes into multiple (>2) leaves. Assume we have an input x that we want to categorize into n cut points $(c_1, c_2, \ldots, c_n)$ that are trainable variables in this context. Then, we calculate the output predictions by applying the forward propagation algorithm of a neural network and comparing those predictions with the actual output values. This helps us reveal and interpret the difference between the predicted and actual ones using a cross-entropy cost

function. Based on this predicted probability, we can decide if the output is either red or green. For instance, the output is green when the value is 1 and red when the value is 0.

We used the cross-entropy loss [75–77] to verify the difference between our prediction and the actual values in Algorithm 2. The cost function defined in Equation (14) is computed. After implementing forward propagation through the NN model, we used back propagation along with the gradient descent algorithm for training our model to determine the derivatives of the loss function with respect to the parameters and updating our parameters ($W_1$, $b_1$, and $W_2$, $b_2$). These steps are repeated until we find the lowest cost or global optimal point.

If we only use the decision tree model, the performance would not be very efficient. However, when the DT learning is integrated with the DNN models, the proposed approach acts as a recursive partitioning for the nonlinearly separable training samples. Particularly in the DTBDNN model, before performing the prediction phase, each node is added to the tree depending on the input samples, which are used to select the logical test at every node. Then, the proposal will decide which model will be used for the data classification. We used TensorFlow to implement our DTBDNN model because it supports "out-of-the-box" GPU acceleration.

## 4. Experimental Results, Performance Evaluations and Discussion

In this section, we present the results of the traditional logistic regression model classification and those of the DTBDNN model (number of hidden layers >= 1) classifications using the datasets in [78]. Then, we validate the efficiency of our proposed model in conjunction with the complex dataset through a subjective test in which we demonstrate that our model would be more stable to learn in the presence of outliers in the dataset with a large number of training samples. By comparing our proposed model to conventional nonlinear data classification, we find that the prediction and classification procedure in our model can converge quickly because it is based on fewer sequences of decisions, with each decision directly dependent on the training samples of the input data.

### 4.1. Dataset and Visualization

We have taken a nonlinearly separable dataset [78], which generates two-class classification. After visualization of the dataset, as depicted in Figure 4a, we demonstrate that it has two classes, represented by the red and green points, in the form of a flower with a color pattern. Specifically, if p = 0, then the data are labeled as red, and if p = 1, then they are labeled as green. The plot shows that the data are not linearly separable. Hence, our goal is to apply DNN classifiers, which are driven by DT, to predict the correct class of data with high accuracy and classify those data using our proposed model.

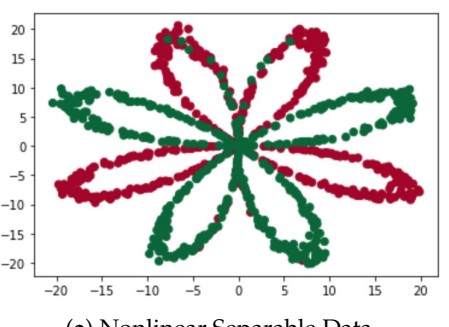 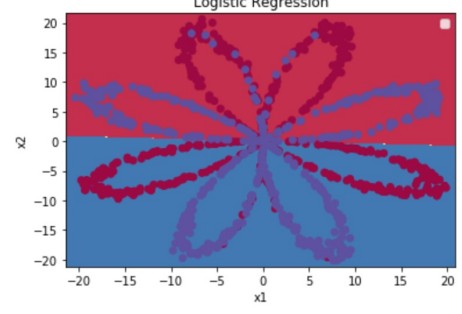

(**a**) Nonlinear Separable Data　　　　　　　　(**b**) Classification using Logistic Regression

**Figure 4.** Classification of nonlinear data using logistic regression model.

### 4.2. Context-Based Logistic Regression Model's Result

Because the obtained training samples in the dataset are not linearly separable, logistic regression (LR) simply draws a straight line to separate the data into two classes, as shown in Figure 4b. Here, we can see that the LR model classifier can only classify 19% of data

points correctly. The result validates that when the data points are not linearly separable, the LR classifier model will not be able to classify these types of data accurately.

We then take another dataset that contains only a linearly separable training sample and visualize the linearly separable data, as illustrated in Figure 5a. When we use these linearly separable data points, as expected, the LR classifier performs the data classification well with much higher accuracy. The decision boundary classification of the linearly separable data for the LR classifier is shown in Figure 5b, where the LR classifier can classify 99% of data points accurately.

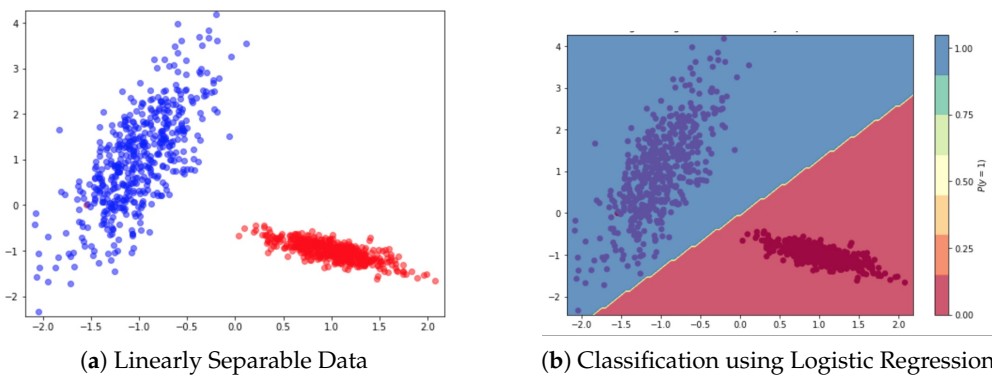

(**a**) Linearly Separable Data　　　　(**b**) Classification using Logistic Regression

**Figure 5.** Classification of linear data using the Logistic Regression model.

Hence, we can deduce that the LR classifier would be the best-fit model for linearly separable data or training samples, while it performs very poorly in the case of nonlinearly separable or complex data classifications.

### 4.3. The Proposed DTBDNN Model's Result

As described in the previous section, the selected parameters are used to predict the classification of the nonlinearly separable planar data group.

For all the training samples (m), we perform 10,000 epochs or iterations, i.e., 10,000 rounds of forward propagation and back propagation, to get the minimum cost. If the cost value is close to zero, then the model performance is said to be converged. We observed that the cost values after every 1000 iterations are decreasing to a close-to-zero value over the iterations, especially when the number of iterations surpasses 1000.

Since the imported dataset input contains nonlinear or planar training samples, we select the NN model using one hidden layer with multiple hidden nodes so that the acquired knowledge from the NN model is expressed and utilized in the DT model. The classification report of the NN model with its performance measurement parameters is shown in Table 1.

**Table 1.** Performance Measurement Parameters of the NN Model.

| Attributes | Precision | Recall | F1-Score | Support |
|------------|-----------|--------|----------|---------|
| 0 | 0.93 | 0.92 | 0.93 | 500 |
| 1 | 0.94 | 0.94 | 0.92 | 500 |
| Micro Avg | 0.94 | 0.93 | 0.93 | 500 |
| Macro Avg | 0.46 | 0.46 | 0.46 | 500 |
| Weighted Avg | 0.93 | 0.93 | 0.93 | 500 |
| Sample Avg | 0.94 | 0.93 | 0.93 | 500 |
| Total | 0.93 | 0.94 | 0.93 | 1000 |

When tuning the hidden layer size, we observe the interesting behavior of the proposed model. Specifically, by increasing the size of the hidden layer (i.e., the number of hidden nodes), we can measure the accuracy of the model and demonstrate its performance in terms of classifying any complex planar data. The accuracy over different numbers of

hidden units in a hidden layer is shown in Table 2. The evaluation results show that the NN model achieves 93% accuracy for nonlinear data classification for hidden node sizes of 4 and above.

**Table 2.** Accuracy over different numbers of hidden nodes in the hidden layer.

| Hidden Layer Size | Accuracy (%) |
|---|---|
| Accuracy for NN Model (No hidden layers) | 93 |
| Accuracy for 1 hidden unit | 71.30 |
| Accuracy for 2 hidden units | 70.899 |
| Accuracy for 3 hidden units | 70.8 |
| Accuracy for 4 hidden units | 93.10 |
| Accuracy for 5 hidden units | 92.10 |
| Accuracy for 20 hidden units | 93.30 |

We now integrate our NN model into the proposed DT classifier. Specifically, the obtained update parameters from the NN model are fed into a set of rules given by the DT algorithm to predict the nonlinear data classification. The nonlinear classified data of our DTBDNN model is plotted in Figure 6a. The DTBDNN model achieves 95% accuracy for nonlinear and linear data classification when the hidden layer size is 4 (Figure 6b). The result shows that DTBDNN can classify the nonlinear or linear dataset's data with much higher accuracy compared to the traditional logistic regression method.

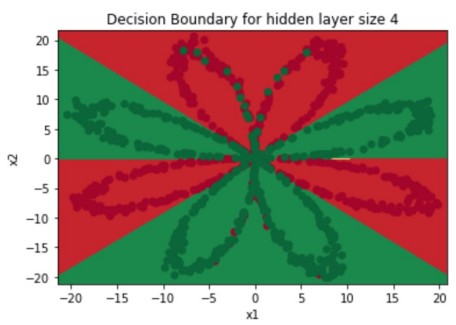
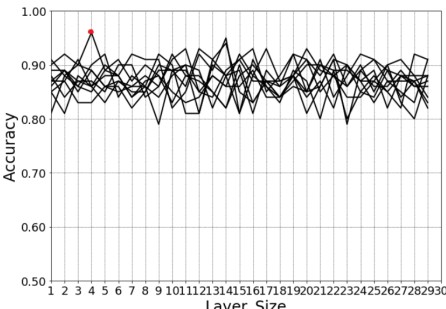

(**a**) Nonlinear data classification (DTBDNN)          (**b**) Accuracy over hidden layer sizes

**Figure 6.** Accuracies of the DTBDNN model to classify nonlinear data.

Figure 7 depicts the convergence of accuracy in the DTBDNN model. The results show that the accuracy in DTBDNN tends to converge when the hidden layer reaches a threshold number, whereas, in the conventional NN model, the accuracy level does not converge for any particular hidden layer size. In fact, accuracy improves when the number of hidden nodes is increased. Typically, in the NN model, when the layer size of a hidden layer is 4, we get 93.1%, and we also get 93.3% accuracy in the case of 20 hidden units. In contrast, in the DTBDNN model, an increase in the size of a hidden layer results in an increase in accuracy, as shown in Figure 7. Furthermore, when the hidden layer size in the DTBDNN model reaches 19, accuracy convergence and maximum accuracy can be achieved, implying that the proposed model achieves 100% accuracy.

The linear/nonlinear data classification for different hidden units in a hidden layer is plotted in Figure 8. Moreover, the confusion matrix for the measurement of the accuracy of the DTBNN model is shown in Figure 9.

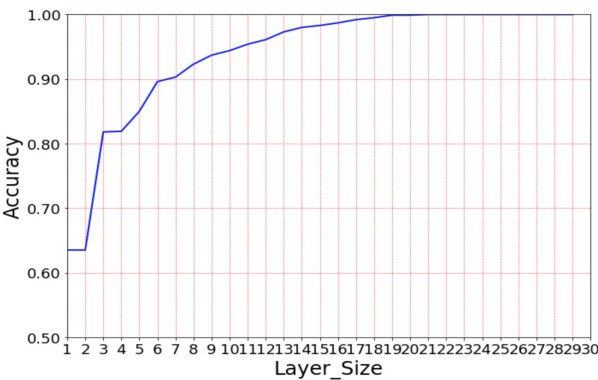

**Figure 7.** Convergence of accuracy over the different layer sizes of the DTBDNN model.

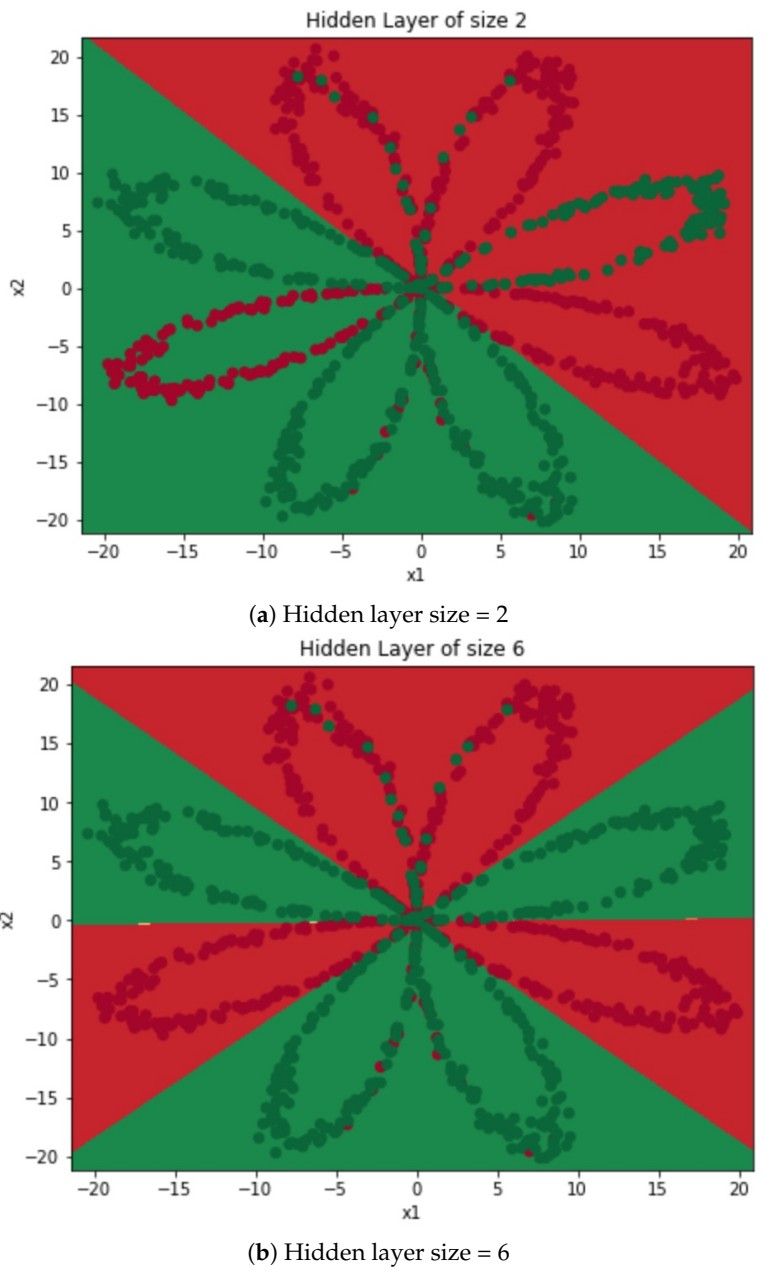

(**a**) Hidden layer size = 2

(**b**) Hidden layer size = 6

**Figure 8.** Decision boundary over different hidden layer sizes of the DTBNN model.

Then, we developed a DTBDNN model with multiple hidden layers and a higher number of hidden units at each layer to classify planar data with maximum accuracy. The accuracy, precision, recall, f1-score, and confusion matrix of the DTBDNN model are shown in Figure 9a. We observe that the DTBDNN model achieves 98% accuracy, and the precision, recall, and f1-score values are higher as well. The reason for the near-maximum accuracy in DTBDNN is the computation of the optimized cost value. Different from the DTBDNN model with only one hidden layer, in which the loss error values are decreased to near zero over iterations to reach the global minimum point for the classification of the planar data, the loss error value in the DTBDNN model is very close to zero after the predefined number of iterations. As a result, the accuracy of the DTBDNN model converges to the peak value quickly, as shown in Figure 9b.

Further, the computation of the cost-effective function and prediction accuracy are shown in Figure 10. The results show that the loss error decreases linearly from nearly 0.6 to 0.1 when the number of iterations is 1600, and the increase in the number of iterations leads to a small value of the loss error, which is almost zero. Hence, the proposed DTBDNN model can achieve a maximum of 100% accuracy in classifying the nonlinear data. The figure shows the computational cost in terms of loss calculation after each epoch, which demonstrates how well our model helps to reach the global minimum point for the classification of the nonlinear data with high accuracy in the DTBDNN model. From this figure, we see that after 1000 iterations, the value of the cost for different numbers of samples does not change.

Next, we take different numbers of training samples and compare the accuracy of the DTBDNN models (number of hidden layers = 1 (DTBNN) vs. number of hidden layers >1 (DTBDNN)), as plotted in Figure 11a. It can be seen that the DTBDNN model has a maximum accuracy of 100%. Moreover, the comparison between the loss error or cost values between DTBNN and DTBDNN over the different numbers of training samples is shown in Figure 11b. The result validates that the loss error values of the DTBNN model are decreasing to nearly zero over iterations but not to zero. For the iterations ranging from 0 to 1000, the loss error decreased linearly from 0.6 to 0.2, and afterward, irrespective of increasing the number of iterations, the loss error values did not decrease much. On the other hand, the loss error values for the DTBDNN model decrease significantly over the iterations, and they are close to zero after the first 1000 iterations. Hence, the DTBDNN model can achieve 100% accuracy, whereas the DTBNN model can only obtain up to 95% accuracy.

The noisy moon nonlinear dataset [78] was then used, as shown in Figure 12a. When fitting this dataset into our DTBDNN model, we achieve 97% accuracy where the cost value is only 0.077227, and the decision boundary for the hidden layer size is depicted in Figure 12b.

Then, we evaluate the application of the DTBDNN model on the noisy Gaussian dataset [78]. The result shows that after fitting the dataset in the DTBDNN model, we achieve 98% accuracy, and the loss error of our model value is minimized at 0.070100 after a pre-specified number of iterations. The corresponding dataset is plotted in Figure 13a, and the decision boundary after fitting the data with the proposed model is shown in Figure 13b.

```
Accuracy : 0.9566666666666667
Precision : 0.9444444444444444
recall : 0.9645390070921985
f1 Score : 0.9543859649122808

[[136   8]
 [  5 151]]
```

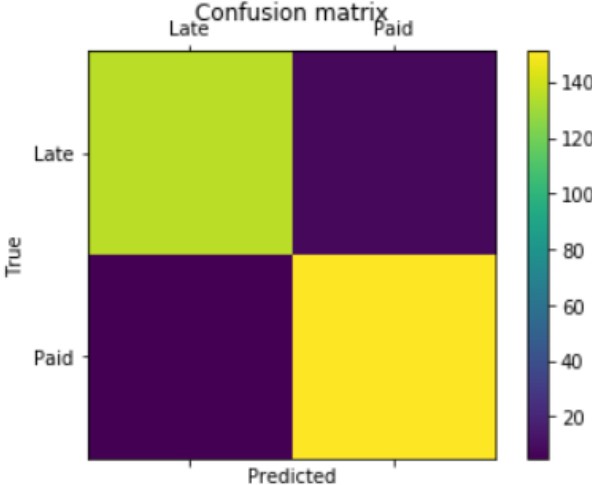

(**a**) Confusion matrix of the DTBNN model

```
Accuracy : 0.98
Precision : 0.9794520547945206
recall : 0.9794520547945206
f1 Score : 0.9794520547945206

[[143   3]
 [  3 151]]
```

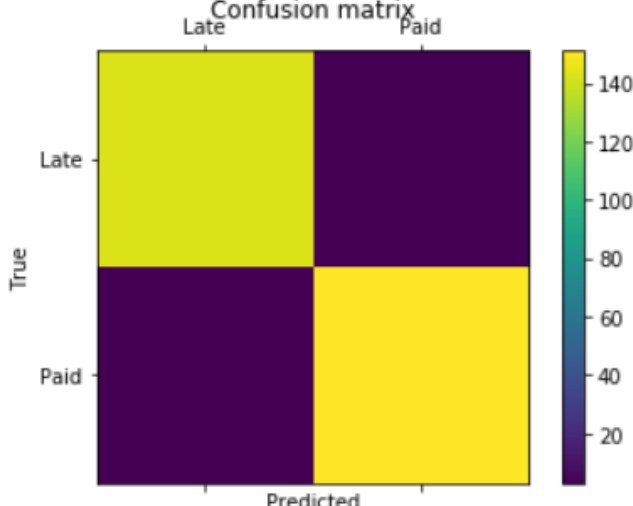

(**b**) Confusion matrix of the DTBDNN model

**Figure 9.** Confusion matrix with accuracy of the DTBNN model and DTBDNN model.

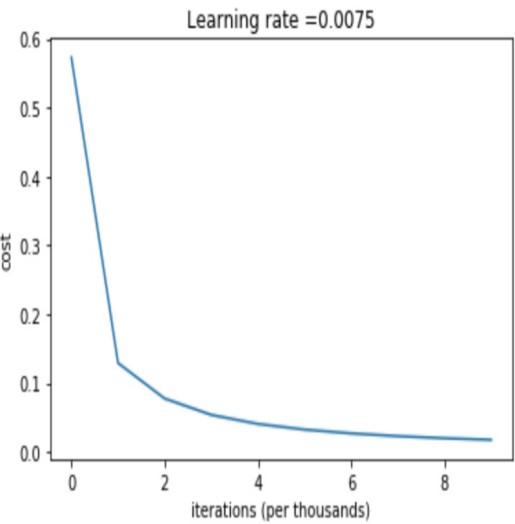

**Figure 10.** Accuracy of the DTBDNN model.

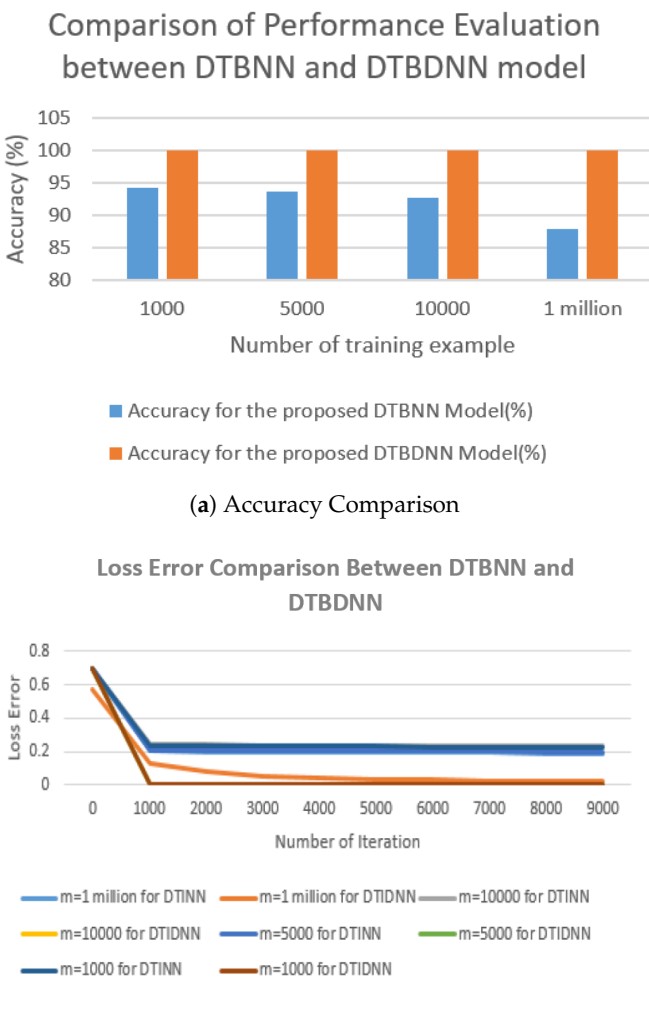

(**a**) Accuracy Comparison

(**b**) Loss Error Comparison

**Figure 11.** Comparison of performance evaluation between DTBNN and DTBDNN models.

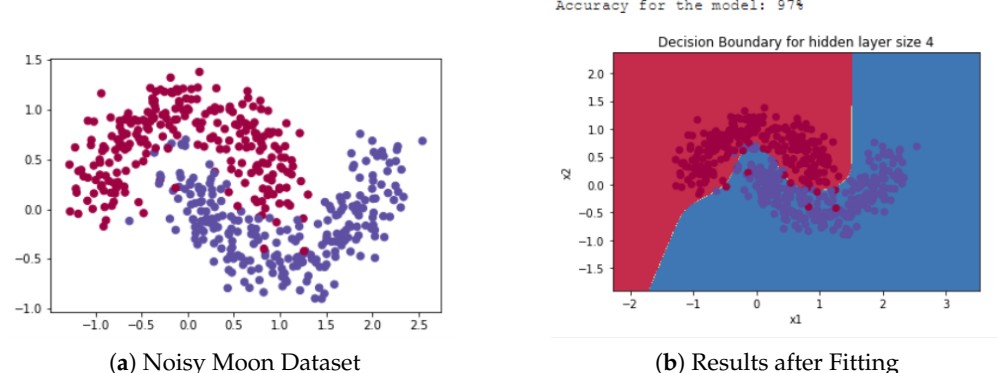

**Figure 12.** Visualization of the noisy moon dataset and results after fitting the DTBDNN model into the noisy moon dataset.

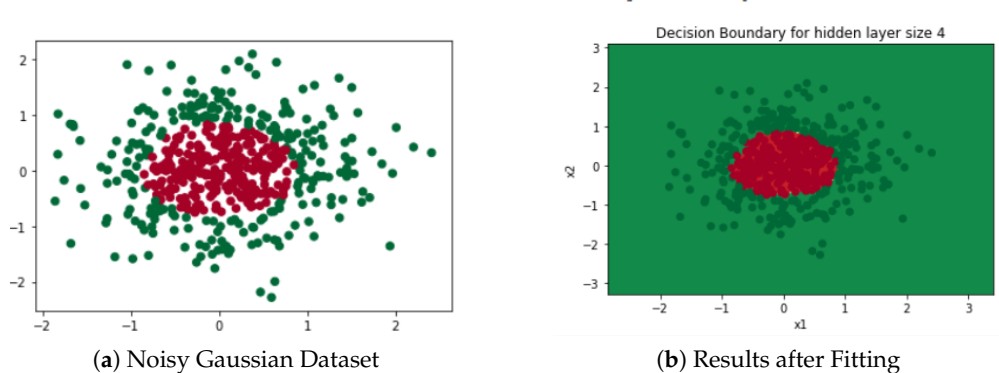

**Figure 13.** Visualization of the noisy Gaussian dataset and results after fitting the DTBDNN model into the noisy Gaussian dataset.

The blobs dataset [78] is another meaningful dataset where multiple circles are plotted on the same surface with different radii. This dataset is also a sample of a nonlinear dataset, and it can be solved by our proposed DTBDNN model with a high accuracy rate of 91%. The training samples of the blobs dataset are shown in Figure 14a, and the decision boundary for the DTBDNN model to classify the blobs dataset's training samples is shown in Figure 14b.

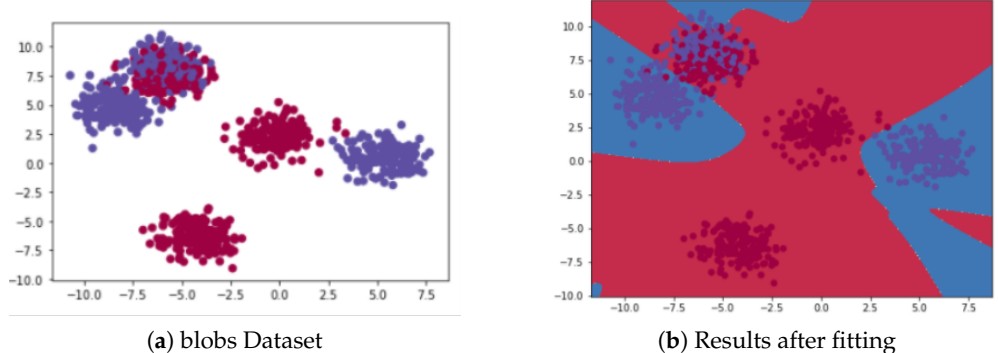

**Figure 14.** Visualization of the blobs dataset and results after fitting the DTBDNN model into the blobs dataset.

Further, Table 2 reveals that our proposed models achieve a maximum accuracy above 90% in different contexts compared to other related and relevant models. The DTBNN and DTBDNN models can then improve the performance of NN and DNN, respectively, by analyzing the loss error function and tuning the hidden layers and output activation functions to optimize the data classification of the separable datasets, especially the nonlinear ones. This proposed framework then enables a feasible and highly efficient approach to training the predictive models for nonlinear data classifications with a wide range of complex nonlinear datasets.

Finally, we have conducted an experiment to evaluate our proposed approach and DT-based algorithm on different datasets collected from UCI. On 10 different datasets gathered from UCI, we evaluate our suggested approach, DTBDNN, versus DT (C4.5 algorithm) versus another state-of-the-art algorithm, ELM-Tree. Table 3 shows the dataset's specifics as well as the test accuracies of DTBDNN, DT, and ELM-Tree. Two of the critical hyperparameter criteria were set to "gini" and "best" for the DT baseline. For the neural network (DTBDNN), we employ a two-hidden-layer architecture with 40 neurons per layer for all datasets. The number of cut points for each feature (also known as the branching factor) is another hyper-parameter in DTBDNN that we set to 1 for all features and datasets. We employ an ensemble of DTBDNN for datasets with more than 12 features, with a total of 10 trees, each of which randomly selects 10 features. The ultimate forecast is provided by majority voting. The DTBDNN is the model that performs the best overall. It is not unexpected that DT performed well, given that these datasets are primarily tabular and have a small feature dimension. Because the hyperparameters in each of these models are adjustable, this is simply an indicative result. It's intriguing that neither model has a clear advantage, though.

Scalability is a problem for induction using decision trees. In other words, the training set of class-labeled tuples stored on disk does not fit in the memory. Or, to put it another way, how scalable is decision tree induction? For comparatively small datasets, the effectiveness of current decision tree algorithms, such as ID3, C4.5, and CART, has been well demonstrated. When these algorithms are used to mine very large real-world databases, efficiency becomes a concern. The limitation of the ground-breaking decision tree algorithms that we have so far covered is that the training tuples must be stored in memory. Very large training sets with millions of tuples are typical in data mining applications. The training data will frequently be too large to fit in memory!

As a result, switching training tuples between main and cache memories makes decision tree construction inefficient. There is a need for more scalable methods that can handle training data that are too big to fit in the memory. Earlier methods of "saving space" included sampling data at each node and discretizing continuous-valued features. However, these methods continue to rely on the notion that the training set may be stored in memory.

Due to mini-batch training in the style of a neural network, DTBDNN scales well with the number of instances. The design, however, has a significant flaw in that it cannot accommodate an increase in the number of features. To avoid this problem with "large" datasets by training a forest with random subspace at the cost of interpretability [52]. Adding numerous trees, each trained on a random subset of characteristics, is what this means. Utilizing the sparsity of the final binning during learning, where the number of non-empty leaves grows far more slowly than the total number of leaves, is a preferable option that avoids the need for an unintelligible forest. However, this makes the otherwise straightforward implementation of DTBDNN a little more complex.

**Table 3.** Testing accuracies of DTBDNN and decision tree (C4.5 algorithm) and ELM-Tree models.

| Dataset | No. of Instances | No. of Features | No. of Classes | DTBDNN | DT | ELM-Tree |
|---|---|---|---|---|---|---|
| Wireless Indoor Localization | 2000 | 7 | 4 | 87.21 | 86.79 | 86.4 |
| OBS-Network | 1075 | 22 | 4 | 96.76 | 95.87 | 96.12 |
| Gime-Me-Some-Credit | 201,669 | 10 | 2 | 97.78 | 91.89 | 95.56 |
| SARS B-cell Epitope Prediction | 14,387 | 13 | 2 | 85.34 | 68.93 | 81.1 |
| Pima Indian Diabetes | 768 | 8 | 2 | 67.23 | 71.56 | 74.48 |
| MAGIC Gamma Telescope | 19,020 | 11 | 2 | 83.56 | 80.76 | 82.58 |
| Waveform Noise | 5000 | 40 | 3 | 75.21 | 69.76 | 75.2 |
| Credit Approval | 690 | 15 | 2 | 81.35 | 83.32 | 81.23 |
| Healthy Older People | 75,128 | 9 | 4 | 97.35 | 95.34 | 96.67 |
| Flight Delay | 1,100,000 | 9 | 2 | 77.89 | 66.67 | 75.34 |

## 5. Conclusions

We created the DTBDNN models in this paper to obtain the most relevant parameters for processing nonlinear data classification by combining the benefits of DT and DNN. In particular, DT extracts the knowledge from the trained DNN models, which are generated from the input data for nonlinear data classification, instead of performing this classification directly from the input data or training samples. A full set of induction algorithms is developed to build and train the DNN model. As a result, the in-depth performance evaluations demonstrate that for classifying any nonlinear data, the proposed models demonstrate a substantial performance improvement compared to the widely used techniques, including decision tree. We then conclude that the proposed model outperformed the relevant state-of-the-art approaches in terms of predicting the nonlinear data classification with the stability of the model and can be used for the realization of efficient prediction to classify any nonlinear or planar data with higher accuracy.

Future employment opportunities are numerous. We want to find out what caused the self-regularization we saw, investigate adding DTBDNN as a module to a traditional convolutional neural network (CNN) feature learner for end-to-end learning of image data, determine whether DTBDNN's whole-tree ADAM-based learning can be used as postprocessing to improve the performance of conventionally greedily trained DTs, and determine whether the various neural-network-based transfer learning approaches can be used to enable transfer learning.

**Author Contributions:** Conceptualization, M.A.; Data curation, T.J.T.; Formal analysis, A.K.P.; Software, M.R.H.; Writing—original draft, T.J.T. and S.B.H.; Writing—review and editing, A.K.P. All authors have read and agreed to the published version of the manuscript.

**Funding:** This research received no external funding.

**Data Availability Statement:** The data used in this study are available upon request.

**Conflicts of Interest:** The authors declare no conflict of interest.

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
