# Peer review of "An Advanced Decision Tree-Based Deep Neural Network in Nonlinear Data Classification"

_technologies, doi:10.3390/technologies11010024_

Round 1
Reviewer 1 Report (Previous Reviewer 1)
The authors responded to my comments and I am recommending that your paper be accepted in its current form.
Author Response
We thank the reviewer for carefully reading our manuscript, pointing out the positive sides of the paper, and appreciating our hard work.

Reviewer 2 Report (New Reviewer)
The paper presents a novel approach for classifying nonlinear data by combining the concepts of Deep Neural Networks (DNNs) and Decision Trees (DTs). The authors propose a Decision Tree-based Neural Network (DTBNN) model and extend it to a Decision Tree-based Deep Neural Network (DTBDNN) model, which utilizes multiple hidden layers in DNNs. The proposed model is tested on various datasets, and the results show that it achieves higher accuracy compared to related and relevant approaches.
Combining DNNs and DTs to classify nonlinear data is straightforward and interesting. DNNs are good at extracting features from complex data, while DT models are simple to decipher and good at handling tabular data. The proposed methods can integrate the strength of both methods.
The paper includes a thorough performance evaluation using different datasets, which demonstrates the effectiveness and feasibility of the proposed approach. The results are clear and easy to understand, and the authors have done an good job in explaining the implications of their findings.
Concerns:
1. Although combining DNNs and DTs is interesting and straightforward and effective, the technical contrition is limited. It would make the paper stronger if it shows its effectiveness in handling image data and other datasets.
2. The resolutions of figures in the paper are not consistent. Some of them are hard to read (e.g Figure 8 - 13)
Author Response
Response to Reviewer 2's Comments and Concerns
Overall Comment:
The paper presents a novel approach for classifying nonlinear data by combining the concepts of Deep Neural Networks (DNNs) and Decision Trees (DTs). The authors propose a Decision Tree-based Neural Network (DTBNN) model and extend it to a Decision Tree-based Deep Neural Network (DTBDNN) model, which utilizes multiple hidden layers in DNNs. The proposed model is tested on various datasets, and the results show that it achieves higher accuracy compared to related and relevant approaches.
Combining DNNs and DTs to classify nonlinear data is straightforward and interesting. DNNs are good at extracting features from complex data, while DT models are simple to decipher and good at handling tabular data. The proposed methods can integrate the strength of both methods.
The paper includes a thorough performance evaluation using different datasets, which demonstrates the effectiveness and feasibility of the proposed approach. The results are clear and easy to understand, and the authors have done an good job in explaining the implications of their findings.
Our Response: We thank the reviewer for carefully reading our manuscript, pointing out the positive sides of the paper, and appreciating our hard work.
Concern 2-1:
Although combining DNNs and DTs is interesting and straightforward and effective, the technical contribution is limited. It would make the paper stronger if it shows its effectiveness in handling image data and other datasets.
Our Response: We thank the reviewer for carefully reading our manuscript and for finding out such important points. To handle image data, we must design a convolutional neural network architecture and integrate decision tree-based learning on top of it.
In this paper, we have proposed a feedforward fully connected neural network to handle tabular data. To handle image data also, we need a convolutional neural network architecture and on top of it we can try to integrate a decision tree model. This requires extensive research to evaluate and obtain fruitful results. We agree with the reviewer that if one model can handle different types of data, then it is more convenient. However, different types of data require specific models (e.g., CNN for image data, RNN for sequence data, etc.) to train. Although recent research shows that, Feedforward fully connected neural networks with attention mechanisms (e.g., BERT, Transformer, etc) can handle different types of data.
We have a plan to do the task mentioned by the reviewer in our future work. We mentioned this as our future work in the last paragraph of the conclusion section (highlighted yellow).
Concern 2-2:
The resolutions of the figures in the paper are not consistent. Some of them are hard to read (e.g Figure 8 - 13)
Our Response: According to the reviewer’s suggestion, we tried to improve the readability of the figures

Reviewer 3 Report (New Reviewer)
The paper needs numerous improvement in order to the reader should follow the paper smoothly
1-The algorithm of Deep NN is not stated well and the reader will get confused. The equations and concepts should be restated and must show the flow of NN in the algorithm.
2-Deep learning algorithm should also be stated separately and the reason why it need to be mixed with NN need to be clarified
3- Why the authors use NN? Why not other approaches?
4- the following paper can be cited in the paper for more clarification of using a robust Database:
EEG-based multi-modal emotion recognition using bag of deep features: An optimal feature selection approach
Author Response
Response to Reviewer 3's Comments
Overall Comment:
The paper needs numerous improvement in order to the reader should follow the paper smoothly.
Our Response: We thank the reviewer for carefully reading our manuscript and for giving useful comments to improve the quality of our manuscript. Our responses to the reviewer’s specific comments are as follows:
Comment 3-1:
1-The algorithm of Deep NN is not stated well and the reader will get confused. The equations and concepts should be restated and must show the flow of NN in the algorithm.
Our Response: We thank the reviewer for carefully reading our manuscript and for finding out such important points. According to the reviewers’ suggestions, we have added necessary equations and concepts required to understand our algorithm clearly in section 3.1.1 (highlighted yellow). Also, we have rewritten the algorithm (Algorithm-2) to reflect the flow of NN.
Comment 3-2:
2-Deep learning algorithm should also be stated separately and the reason why it need to be mixed with NN need to be clarified
Our Response: We would like to thank the reviewer for their constructive comments. We have taken into consideration the reviewer’s comment and added a separate generalized deep learning algorithm (Algorithm-1 (highlighted yellow)) to make the concepts clear for the reader. When the number of hidden layers is 1, we call it NN and when the number of hidden layers is greater than 1, we call it DNN. However, since this creates confusion, we have combined this concept into one (Deep Neural Network (DNN) and distinguishes by the number of layers when we present the experimental results).
Comment 3-3:
3- Why the authors use NN? Why not other approaches?
Our Response: According to the reviewer’s concern, we have discussed about this issue in section 1, paragraph 4 and 6 (highlighted yellow).
Comment 3-4:
4- the following paper can be cited in the paper for more clarification of using a robust Database:
EEG-based multi-modal emotion recognition using bag of deep features: An optimal feature selection approach
Our Response: According to the reviewer’s suggestions, we have added the suggested paper as a reference.

Round 2
Reviewer 3 Report (New Reviewer)
The authors did their best to revise the paper and it is now acceptable \
This manuscript is a resubmission of an earlier submission. The following is a list of the peer review reports and author responses from that submission.
Round 1
Reviewer 1 Report
The paper is very well structured. The Introduction section is good, in this section the authors present clearly the objectives and the main contributions of the study. The authors provided sufficient background and include relevant references. The method is adequately described. The results are clearly presented. The conclusions are supported by the results.
Author Response
We thank the reviewer for carefully reading our manuscript, pointing out the positive sides of the paper, and appreciating our hard work.

Reviewer 2 Report
The authors proposed DTBNN and DTBDNN algorithms to induce hybrid machine learning models to tackle problems based on non-linear scenarios, overcoming the current state-of-the-art algorithms. The authors claimed that both proposed algorithms, taking advantage of Decision Trees and Neural Network characteristics are able to create highly predictive solutions.
Unfortunately, the current paper version describes biased experiments that do not support the claimed contribution. The main suggestions to reach the expected contribution, in my point of view, are:
1- The authors need to well-ground several affirmations, such as “the state-of-the-art in non-linear classification”, the selected dataset, etc. Also, one of the most important affirmations, particularly about the results of a DT being comparable (and also overcoming) a NN was based on a single paper from arXiv. It is expected to read robust research and discussion about this. In fact, I am not convinced that DT is able to overcome the performance of NN in several different classification tasks.
2- The authors need to emphasize the advantages of Decision Trees and Neural Networks that motivate the hybrid proposal. In this current version, there is just some naive information. Also, how the current solutions (both models) deal with the main drawback of DTs and NN.
3- The authors need to explicitly present the scenario, goals, and contributions of the current work. These pieces of information need to be used in the experiments. Otherwise, it is not possible to prove the expected contribution. Some important points:
- Binary or Multi-class scenario;
- Balanced or unbalanced datasets;
- The non-linearity level;
- Scalability issues;
- Explainability issues. DT’s contribution used to support XAI perspectives but included a “black-box” NN the XAI aspect could be compromised
- Tuning procedures need to be discussed, considering the current proposal and the state-of-the-art.
- Fair comparison to other algorithms, it is mandatory using more than 10 benchmark datasets;
- What are the state-of-the-art algorithms employed in the experiments?
- Statistically evaluating the significance of the performance.
4- Architectural Overview needs to be revisited, for example, the proposal presents two “prediction activities”. Is it the proposed approach or the experimental setting?
5- The images need to be improved, they are distorted;
6- The algorithm does not follow the expected quality, the inputs are not clearly defined, and the limitations, output, and sequence of functions do not match the expected standard. The pseudo-code needs to represent the instructions and ideas and does not represent the source implemented code.
Author Response
Overall Comment of the reviewer:
The authors proposed DTBNN and DTBDNN algorithms to induce hybrid machine learning models to tackle problems based on non-linear scenarios, overcoming the current state-of-the-art algorithms. The authors claimed that both proposed algorithms, taking advantage of Decision Trees and Neural Network characteristics are able to create highly predictive solutions.
Our Response:
We thank the reviewer for carefully reading our manuscript and for giving useful comments to improve the quality of our manuscript. Our responses to the reviewer’s specific comments are as follows:
Comment 2-1:
The authors need to well-ground several affirmations, such as “the state-of-the-art in non-linear classification”, the selected dataset, etc. Also, one of the most important affirmations, particularly about the results of a DT being comparable (and also overcoming) a NN was based on a single paper from arXiv. It is expected to read robust research and discussion about this. In fact, I am not convinced that DT is able to overcome the performance of NN in several different classification tasks.
Our Response: According to the reviewer’s concern, we have added Table 3 in section 4 and the corresponding discussions are added in last paragraph of section 4.
Comment 2-2:
The authors need to emphasize the advantages of Decision Trees and Neural Networks that motivate the hybrid proposal. In this current version, there is just some naive information. Also, how the current solutions (both models) deal with the main drawback of DTs and NN.
Our Response: We would like to thank the reviewer for such constructive comments. According to the reviewer’s suggestion, we have addressed this issue and added 2nd and 3rd paragraph in the introduction section that discusses the reviewers concerns.
Comment 2-3:
The authors need to explicitly present the scenario, goals, and contributions of the current work. These pieces of information need to be used in the experiments. Otherwise, it is not possible to prove the expected contribution. Some important points:
- Binary or Multi-class scenario;
- Balanced or unbalanced datasets;
- The non-linearity level;
- Scalability issues;
- Explainability issues. DT’s contribution used to support XAI perspectives but included a “black-box” NN
the XAI aspect could be compromised
- Tuning procedures need to be discussed, considering the current proposal and the state-of-the-art.
- Fair comparison to other algorithms, it is mandatory using more than 10 benchmark datasets;
- What are the state-of-the-art algorithms employed in the experiments?
- Statistically evaluating the significance of the performance
Our Response: According to the reviewer’s concern, we have added Table 3 in section 4 and the corresponding discussions are added in the last paragraph of section 4.
Comment 2-4:
Architectural Overview needs to be revisited, for example, the proposal presents two “prediction activities”. Is it the proposed approach or the experimental setting?
Our Response: We would like to thank the reviewer for constructive comments. We have taken into consideration the reviewer’s comment and modified the architectural overview accordingly that can be found in Fig.1 and the explanation is given in section 3.1, paragraph 3.
Comment 2-5:
The images need to be improved, they are distorted;
Our Response: Suggested modification has been done.
Comment 2-6:
The algorithm does not follow the expected quality, the inputs are not clearly defined, and the limitations, output, and sequence of functions do not match the expected standard. The pseudo-code needs to represent the instructions and ideas and does not represent the source implemented code.
Our Response: We would like to thank the reviewer for the valuable suggestions. However, we partially disagree with the reviewer. In our algorithm, we have clearly defined the input, output, and processing sections. Each section clearly defines the instructions necessary to execute our idea.

Reviewer 3 Report
The paper deals with important task. The authors developed and extend a Decision Tree-Based Neural Network (DTBNN).
The idea is very interesting. The paper has great practical value.
It has a logical structure all necessary sections. The paper is technically sound. The experimental section is very good.
The proposed approach is logical, results are clear.
In general paper is very good.
Suggestions:
1. The introduction and related works section should be extended using 10.5815/ijmecs.2021.03.02, 10.5815/ijisa.2017.09.04, 10.5815/ijisa.2017.10.07
2. Why the authors chosed Logistic regression but not SVR with nonlinear kernels as a model for comparison.
3. Many figures has a bad quality. Please improve it
4. The conclusion section should be extended using: 1) numerical results obtained in the paper; 2) limitations of the proposed approach; 3) prospects for future research.
Author Response
Overall Comment:
The paper deals with important task. The authors developed and extend a Decision Tree-Based Neural Network (DTBNN). The idea is very interesting. The paper has great practical value. It has a logical structure all necessary sections. The paper is technically sound. The experimental section is very good. The proposed approach is logical, results are clear. In general paper is very good.
Our Response: We thank the reviewer for carefully reading our manuscript, appreciating our hard work, and giving useful comments to improve the quality of our manuscript. Our responses to the reviewer’s specific comments are as follows:
Comment 3-1:
The introduction and related works section should be extended using 10.5815/ijmecs.2021.03.02, 10.5815/ijisa.2017.09.04, 10.5815/ijisa.2017.10.07
Our Response: We thank the reviewer for carefully reading our manuscript and for giving valuable comments. According to the reviewer’s suggestions, we have extended the introduction section (first paragraph of the revised paper) and added the appropriate references to the paper suggested.
Comment 3-2:
Why the authors chose Logistic regression but not SVR with nonlinear kernels as a model for comparison.
Our Response: We tried to show that for non-linear dataset, logistic regression is not appropriate to choose.
Comment 3-3:
Many figures have a bad quality. Please improve it
Our Response: According to the reviewer’s concern, we have improved the quality of the figure.
Comment 3-4:
The conclusion section should be extended using: 1) numerical results obtained in the paper; 2) limitations of the proposed approach; 3) prospects for future research.
Our Response: According to the reviewer’s suggestions, we have rewritten the conclusion to reflect the reviewer’s suggestion.

Reviewer 4 Report
This paper presents a model which uses a NN to learn splits for a DT. The paper is not that well written, although there seems to be much work behind it.
I fail to understand what is analysed in the Related Work section. It is merely a listing of some methods/applications of DNNs, some hybridized with DTs, ensemble learning, but it doesn't seem to lead anywhere.
The method proposed is not at all clearly described. The paper fails to appropriately explain how the architecture looks like, what is the role of the NN neurons in the DT model, and how the NN weights affect the DT model. The diagram in figure 1 is very vague and non-informative. The model is restricted to two class problems. How would it extend to multiclass?
Since the evaluations are performed on UCI data, all results should clearly state which dataset they refer to. Otherwise, they are meaningless. Moreover, a DT model would be a better fit as a baseline than LR in this case, since the proposed solution is of a DT based on a NN.
The paper contains many formulations which need revisiting. I have outlined some below:
- "it is crucial but difficult to develop a particular data classification model from a new and unforeseen dataset." - in what sense difficult? It is vague. difficulty to build the model does not come from the lack of transparency necessarily, but the difficulty in using the model might come from that.
- "We first developed a Decision Tree-Based Neural Network (DTBNN) 6 model Next, we extend..". missing . between sentences
- "our proposal achieves the optimal trainable weights and bias to build an efficient model for the nonlinear data classification." - optimal training weights means that you guarantee reaching the absolute minima for the loss. Given that the loss space is not convex, how can you guarantee that?
- "by achieving an accuracy of 95% in the 12 DTBNN model and 98% for the DTBDNN model." - absolute accuracy measures without stating the dataset they were obtained on are meaningless. E.g. on the MNIST dataset, a 95% accuracy is considered weak.
- "In deep learning, Deep Neural Network (DNN)," - "In deep learning, a Deep Neural Network (DNN),"
-"In general, NN consists of the input layer, " - "In general, a NN consists of the input layer, "
- " In general, logistic regression is mostly used for linearly separable data since it gives a lower classification error than any other model [24]. " - "training or validation error??"
- " logistic regression demonstrates a relatively high classification error rate." - poor choice of phrasing
- "decision tree-based neural network (DTBNN) and decision tree-based deep neural network (DTBDNN), respectively." DL does not only refer to number of hidden layers...and I do not agree that a 1-hidden layer NN is shallow, whereas a 2-hidden layer NN is deep.
-"the performance would be not very efficient," - performance is either good/bad, high/low; efficiency relates to the method directly, depending on the level of performance achieved by it.
- etc.
Author Response
Overall Comment:
This paper presents a model which uses a NN to learn splits for a DT. The paper is not that well written, although there seems to be much work behind it.
Our Response: We thank the reviewer for carefully reading our manuscript and for giving useful comments to improve the quality of our manuscript. Our responses to the reviewer’s specific comments are as follows:
Comment 4-1:
I fail to understand what is analyzed in the Related Work section. It is merely a listing of some methods/applications of DNNs, some hybridized with DTs, ensemble learning, but it doesn't seem to lead anywhere.
Our Response: We thank the reviewer for carefully reading our manuscript and for finding out such important points. According to the reviewers’ suggestions, we have added paragraphs 2, 3, 4, and 5 in the related work section in the revised paper and removed paragraphs 2 and 4 from the previously submitted paper. In the revised paper, the newly added paragraph describes the related work that is useful for understanding our proposed work.
Comment 4-2:
The method proposed is not at all clearly described. The paper fails to appropriately explain how the architecture looks like, what is the role of the NN neurons in the DT model, and how the NN weights affect the DT model. The diagram in figure 1 is very vague and non-informative. The model is restricted to two class problems. How would it extend to multiclass?
Our Response: We would like to thank the reviewer for their constructive comments. We have taken into consideration the reviewer’s comment and modified the architectural overview accordingly which can be found in Fig.1 and the explanation is given in section 3.1, paragraph 3.
Comment 4-3:
Since the evaluations are performed on UCI data, all results should clearly state which dataset they refer to. Otherwise, they are meaningless. Moreover, a DT model would be a better fit as a baseline than LR in this case, since the proposed solution is of a DT based on a NN.
Our Response: According to the reviewer’s concern, we have added Table 3 in section 4 and the corresponding discussions are added in the last paragraph of section 4.
Comment 4-4:
The paper contains many formulations which need revisiting. I have outlined some below:
- "it is crucial but difficult to develop a particular data classification model from a new and unforeseen dataset." - in what sense difficult? It is vague. difficulty to build the model does not come from the lack of transparency necessarily, but the difficulty in using the model might come from that.
- "We first developed a Decision Tree-Based Neural Network (DTBNN) 6 model Next, we extend..". missing . between sentences
- "our proposal achieves the optimal trainable weights and bias to build an efficient model for the nonlinear data classification." - optimal training weights means that you guarantee reaching the absolute minima for the loss. Given that the loss space is not convex, how can you guarantee that?
- "by achieving an accuracy of 95% in the 12 DTBNN model and 98% for the DTBDNN model." – absolute accuracy measures without stating the dataset they were obtained on are meaningless. E.g. on the MNIST dataset, a 95% accuracy is considered weak.
- "In deep learning, Deep Neural Network (DNN)," - "In deep learning, a Deep Neural Network (DNN),"
-"In general, NN consists of the input layer, " - "In general, a NN consists of the input layer, "
- " In general, logistic regression is mostly used for linearly separable data since it gives a lower classification error than any other model [24]. " - "training or validation error??"
- " logistic regression demonstrates a relatively high classification error rate." - poor choice of phrasing
- "decision tree-based neural network (DTBNN) and decision tree-based deep neural network (DTBDNN), respectively." DL does not only refer to number of hidden layers...and I do not agree that a 1-hidden layer NN is shallow, whereas a 2-hidden layer NN is deep.
-"the performance would be not very efficient," - performance is either good/bad, high/low; efficiency relates to the method directly, depending on the level of performance achieved by it.
- etc.
Our Response: According to the reviewer’s suggestions, all the modifications are done.

Round 2
Reviewer 2 Report
Thank you for considering my suggestions and recommendations, however, they were covered partially. The current version lacks experiment rigor and discussions.
The authors added a comparison to C4.5, but exploring the wide range of Decision Tree algorithms is mandatory, particularly the state-of-the-art algorithms. Also, the authors need to discuss the mentioned topics regarding cardinality, balance, and several important points to draw the advantages and disadvantages of the current proposal.
Considering these limitations, I’m not convinced that the current contribution supports the claimed importance.
Author Response
Reviewer Comment:
Thank you for considering my suggestions and recommendations, however, they were covered partially. The current version lacks experiment rigor and discussions.
The authors added a comparison to C4.5, but exploring the wide range of Decision Tree algorithms is mandatory, particularly the state-of-the-art algorithms. Also, the authors need to discuss the mentioned topics regarding cardinality, balance, and several important points to draw the advantages and disadvantages of the current proposal.
Considering these limitations, I’m not convinced that the current contribution supports the claimed importance.
Our Response:
We partially disagree with the reviewer's comment. First, we have compared a performance evaluation with our proposed work with a decision tree-based algorithm. We chose the C4.5 algorithm to compare our work with several algorithms. Different datasets are used to evaluate the proposed work and C4.5. From the experimental results, we concluded that in most of the cases our proposed work achieves better results as compared to C4.5. And in some cases where the dataset is small, C4.5 outperforms our proposed work. So we come to the conclusion that for the larger dataset, our proposed model has a clear advantage. We could have added other decision tree-based state-of-the-art algorithms. But we believe that it will not make any difference.
Overall, we tried our best to address the major concerns of all the reviewers and revised our paper to meet the standard of the journal.
Reviewer 3 Report
The authors addressed all my comments. The paper can be accepted in current form
Author Response
Thank you very much for appreciating our hard work.